# Antimicrobial and Antibiofilm Properties of Latvian Honey against Causative Agents of Wound Infections

**DOI:** 10.3390/antibiotics12050816

**Published:** 2023-04-26

**Authors:** Ingus Skadiņš, Krišs Dāvids Labsvārds, Andra Grava, Jhaleh Amirian, Laura Elīna Tomsone, Jānis Ruško, Arturs Viksna, Dace Bandere, Agnese Brangule

**Affiliations:** 1Baltic Biomaterials Centre of Excellence, Headquarters at Riga Technical University, LV-1658 Riga, Latvia; ingus.skadins@rsu.lv (I.S.); dace.bandere@rsu.lv (D.B.); 2Department of Biology and Microbiology, Riga Stradiņš University, LV-1007 Riga, Latvia; 3Faculty of Chemistry, University of Latvia, LV-1004 Riga, Latvia; kriss_davids.labsvards@lu.lv (K.D.L.); arturs.viksna@lu.lv (A.V.); 4Institute of Food Safety, Animal Health and Environment “BIOR”, LV-1076 Riga, Latvia; 5Rudolfs Cimdins Riga Biomaterials Innovations and Development Centre of RTU, Faculty of Materials Science and Applied Chemistry, Institute of General Chemical Engineering, Riga Technical University, LV-1007 Riga, Latvia; 6Department of Pharmaceutical Chemistry, Riga Stradiņš University, LV-1007 Riga, Latvia

**Keywords:** honey, antimicrobial activity, biofilms, polyphenolic profile

## Abstract

Honey is widely used in traditional medicine and modern wound healing biomaterial research as a broad-spectrum antimicrobial, anti-inflammatory and antioxidant agent. The study’s objectives were to evaluate the antibacterial activity and polyphenolic profiles of 40 monofloral honey samples collected from beekeepers in the territory of Latvia. The antimicrobial and antifungal activity of Latvian honey samples were compared with commercial Manuka honey and the honey analogue sugar solutions–carbohydrate mixture and tested against *Escherichia coli*, *Pseudomonas aeruginosa*, *Staphylococcus aureus*, clinical isolates *Extended-Spectrum Beta-Lactamases* produced *Escherichia coli*, *Methicillin-resistant Staphylococcus aureus* and *Candida albicans*. Antimicrobial activity was evaluated with the well-diffusion method (80% honey solution *w*/*v*) and microdilution method. The honey samples with the highest antimicrobial potential were tested to prevent biofilm development and activity against a preformed biofilm. The principal component analysis of the antimicrobial properties of honey samples vs. polyphenolic profile was performed. Eleven honey samples exhibited antibacterial activity to all investigated bacteria. The antibacterial effect of the samples was most significant on the Gram-positive bacteria compared to the studied Gram-negative bacteria. Latvian honey presents promising potential for use in wound healing biomaterials, opening the possibility of achieving long-term antibacterial effects.

## 1. Introduction

Antibacterial agents are essential in the management and treatment of chronic wounds [1,2,3]. However, although antibiotics are available to treat skin infections, overuse can cause some problems, such as developing bacterial resistance and creating adverse side effects [4]. Therefore, research for new alternatives of antibiotics or antibacterial agent combinations leads to studying ingredients in plants and natural products such as honey [5]. Several studies recommend using honey directly or with antibiotics because it increases the antimicrobial effect and creates synergistic or additive interactions against bacterial biofilms [6,7]. Using honey alone or together with antibiotics could reduce the use of antibiotics, minimize adverse antibiotic reactions and increase the effectiveness of treatment [8]. Although the main constituents of honey are reducing carbohydrates, the composition of honey is very complex, and it contains about 200 substances [9].

The high sugar content and viscosity of honey have been shown to play a crucial role in inhibiting microbial growth and preventing the formation of biofilms. Furthermore, the biofilm matrix is composed of polysaccharides, and recent studies have shown that sugar molecules are also utilized as chemical messengers between bacterial species within the biofilm structure [10].

In addition, honey is a rich natural source of phenolic compounds, which can be used to accelerate the wound healing process and as markers for determining botanical origin [11]. Polyphenols are a group of natural compounds that are widely distributed in plants and have been found to possess antibacterial properties against a range of Gram-positive and Gram-negative bacteria. Various mechanisms of the antimicrobial action of polyphenols have been described in the literature, for e.g., they disrupt the bacterial cell membrane (pore formation, disintegration of membrane proteins, cell wall disruption, modifying membrane potential), affect cytoplasm (leaking cell components, cytoplasm acidification, chelation of metal ions) and disturb functions (DNA/RNA/protein synthesis, modulate a cellular redox response through a proline-linked pentose phosphate pathway, inactivation of enzymes, loss of biofilm formation) [12,13,14]. Similar to honey, polyphenols also have also been shown to enhance the activity of antibiotics against biofilms of several bacterial species, including *Pseudomonas aeruginosa* (PA) and *Staphylococcus aureus* (SA) [15].

Scientific papers have extensively studied the properties and applications of Manuka honey, which is produced from the *Leptospermum scoparium* tree in New Zealand [16,17,18]. Manuka honey, with its special antibacterial and anti-inflammatory properties, found a medical application in wound treatment and is incorporated into biomaterials for medical implants [19,20,21,22]. Studies mainly highlight the beneficial activity of Manuka honey against various bacteria, such as *Staphylococcus aureus* (SA) and *Escherichia coli* (EC), as well as against biofilm formation [17,23]. The high antimicrobial activity of honey is explained by the specific composition (flavonoids, phenolic acids, methylglyoxal) and the low pH value, between 3.2–4.5 [24]. Latvia has an extensive tradition of honey production and use. Although honey has been used for medical purposes in Latvia for centuries, the analysis of polyphenols and other compounds has only started recently [25,26]. Therefore, it is important to investigate monofloral honey samples collected in Latvia, their polyphenolic profile, their antimicrobial activity against Gram-positive and Gram-negative bacteria and their antifungal properties.

Gram-negative bacterial strains such as *Escherichia coli* (EC) and *Pseudomonas aeruginosa* (PA) and their antibiotic-resistant variants such as *Extended-Spectrum Beta-Lactamases* (ES) can form biofilms, which is a challenge in modern medicine due to the spread of resistance and the formation of biofilms. Gram-positive bacteria such as *Staphylococcus aureus* (SA) and the antibiotic-resistant form of *Methicillin-resistant Staphylococcus aureus* (MR) also have a strong ability to form biofilms [27]. At the same time, all these bacteria are the most common causative agents of wound infections, which can be acquired in hospitals or via community spreading [28]. One of the threatening fungal causative agents for burn patients is yeasts, i.e., *Candida albicans*, which, as an opportunistic causative agent, can cause wound infections for burn patients [29].

Our recent study aimed to evaluate the antibacterial activity and polyphenolic profiles of 40 monofloral honey samples collected from beekeepers in the territory of Latvia.

## 2. Results

### 2.1. Polyphenol Profile of Latvia Common Monofloral Honey

The mean concentrations (μg/kg) of 17 different polyphenols were studied in 40 honey samples and compared to their botanical origins. The concentrations of one plant hormone (abscisic acid) and seven phenolic acids (p-hydroxybenzoic acid, p-coumaric acid, 3,4-dihydroxybenzoic acid, ferulic acid, syringic acid, chlorogenic acid, gallic acid) were summarized in Table 1; eight flavonoids (kaempferol, rutin, luteolin, genistein, galangin, acacetin, isovitexin, formononetin) and one water-soluble vitamin (pantothenic acid) were summarized in Table 2.

Fourteen polyphenols were detected in all samples: 3,4-dihydroxybenzoic acid, abscisic acid, acacetin, chlorogenic acid, ferulic acid, galangin, gallic acid, genistein, kaempferol, luteolin, p-coumaric acid, p-hydroxybenzoic acid, pantothenic acid and synergic acid. In addition, formononetin, isovitexin and rutin were determined in 13, 12 and 32 samples out of 40, respectively.

Regarding the botanical origin, formononetin was not detected in any buckwheat honey and apiaceae honey samples but was found in all clover samples. Isovitexin was not found in any sample of linden flower honey and apiaceae honey.

The highest levels of polyphenols were observed for abscisic acid, p-hydroxybenzoic acid, p-coumaric acid and ferulic acid. The highest concentrations of abscisic acid were observed for all willow honey, where four samples had concentrations from 10,326 to 16,117 μg/kg, and one had as much as 84,712 μg/kg. The highest concentrations of p-hydroxybenzoic acid were detected in buckwheat honey; the concentrations ranged for all samples from 13,990–19,823 μg/kg, and one sample was 7219 μg/kg. Similarly, high p-hydroxybenzoic acid concentrations were observed in only two samples of clover honey, Clo_4 and Clo_6. Like p-hydroxybenzoic acid, p-coumaric acid also had the highest values found in buckwheat honey samples, and the minimal and maximal concentrations were 3811 and 6977 μg/kg, respectively. Ferulic acid was found in five honey samples with relatively lower concentrations, and they were three linden flower samples, Lin_3, Lin_5 and Lin_7, with concentrations of 926 μg/kg, 964 μg/kg and 962 μg/kg, respectively; one buckwheat honey, Buck_4, had a concentration of 946 μg/kg; and one willow honey Wil_4 had a concentration of 744 μg/kg. The highest rutin concentrations were observed for buckwheat honey samples between 347–816 μg/kg. However, the samples with particularly high rutin concentrations should be mentioned; they were Api_1, Wil_7 and Rap_6, which had rutin concentrations of 1076, 1297 and 2466 μg/kg, respectively.

According to the botanical origin, the most polyphenolic compounds were observed in buckwheat honey, the concentrations of which were increased: 3,4-dihydroxybenzoic acid, chlorogenic, p-coumaric acid, p-hydroxybenzoic acid and rutin.

### 2.2. Effect of Honey on Bacterial Growth

To study the antibacterial and antifungal properties of honey, in this project, we used the well-diffusion, MIC, MBC and MFC determinations with the microdilution method, as well as the study of the prevention of biofilm development and activity of honey samples against preformed biofilm.

#### 2.2.1. Well-Diffusion Method

In the present study, 32 honey samples of 40 showed activity against EC; 23 against ES, 29 against PA, 29 against SA and 33 against MR. The honey analogue sugar solutions did not show any inhibition on the bacteria and fungi. Against CA, the antifungal activity was practically not detected. An example against the *Methicillin-Resistant Staphylococcus aureus* (MR) from the test results after the well-diffusion method is presented in Figure 1.

Eleven honey samples (Api_1; Buck_1, Buck_4, Clo_6, Clo_7, Hea_1, Hor_1, Phi_1, Rap_4, Wil_6 and Wil_7) exibited antibacterial activity to all investigated bacteria. However, four honey samples (Api_2, Wil_1, Wil_2 and Wil_5) showed no activity on bacteria except on PA.

Compared to Manuka honey, two honey samples exibited higher inhibition zones on ES; two on EC, eleven honey samples on SA, ten on PA and ten on MR.

The antibacterial effect of the samples was most significant on the Gram-positive SA and MR bacteria compared to the investigated Gram-negative bacteria. The highest values of the inhibition zones for SA and MR correspondingly were 22 mm and 26 mm compared to EC—17 mm, ES—12 mm and PA—13 mm. Against SA bacteria, seven samples (Buck_4, Buck_5, Buck_2, Api_1, Rap_5, Wil_6 and Lin_7) showed very high inhibition zones (17–22 mm), and against MR 12 samples (Buck_2, Clo_7, Clo_6, Clo_4, Buck_1, Buck_4, Hea_1, Wil_6, Api_1, Buck_5, Rap_5 and Lin_7) exhibited values (15–26 mm). The bar charts in Figure 2 and Appendix A disclose the antibacterial effect of the honey samples.

#### 2.2.2. Minimum Inhibitory Concentration (MIC), Minimum Bactericidal Concentration (MBC) and Minimums Fungicidal Concentration (MFC)

The lowest MIC value (10%) was observed for ten honey samples (Api_1, Buck_1, Buck_2, Buck_4, Buck_5, Clo_7, Lin_7, Rap_5, Wil_6 and Man) against EC. Five of ten honey samples (Api_1, Buck_4, Lin_7, Rap_5, Man) showed the lowest MBC value of 10% against EC. Comparing the lowest MIC values between EC and ES, it can be seen that only three honey samples (Lin_7, Wil_6 and Man) showed a MIC of 10%, and from them, only Lin_7 and Wil_6 samples showed a MBC of 10% against ES.

Much better activity was observed against Gram-positive bacteria compared to Gram-negative bacteria; respectively, the lowest MIC value of 2.5% and a MBC value of 2.5% were seen for Lin_7 against SA and MR. Eight honey samples (Api_1, Buck_1, Buck_2, Buck_4, Buck_5, Rap_5, Wil_6 and Man) showed a MIC value of 5%, and seven honey samples (Buck_3, Clo_4, Clo_6, Clo_7, Hea_1, Hea_3 and Hor_1) showed a MIC value of 10% against SA. All honey samples (except Man), which showed a MIC value of 5%, also showed a MBC value of 5% against SA. Ten honey samples (Api_1, Buck_1, Buck_2, Buck_3, Buck_4, Buck_5, Clo_4, Rap_5, Wil_6 and Man) showed a MIC value of 5% against MR, and six of them (Api_1, Buck_2, Buck_3, Buck_5, Rap_5 and Wil_6) also showed MBC values of 5%.

Against Gram-negative, PA has seen the same relevance as it had with EC and ES. The lowest MIC values were 10% for seven honey samples (Api_1, Buck_2, Buck_4, Buck_5, Lin_7, Rap_5 and Wil_6), and all MBC values were 10%, except for Rap_5 and Buck_2—the MBC value was 20% for these samples.

The lowest MIC value of 20% was observed against CA for Buck_4 and Lin_7, but the MFC for both samples was 40% against CA. In general, the activity of honey samples against CA was observed poorly (Table 3).

#### 2.2.3. Effect of Honey on Biofilms

##### Antibiofilm Activity—Prevention of Biofilm Development

Sixteen honey samples (Api_1, Buck_1, Buck_2, Buck_3, Buck_4, Buck_5, Clo_4, Clo_6, Clo_7, Hea_1, Hea_3, Hor_1, Lin_7, Rap_5, Wil_6 and Manuka) which showed the best antibacterial properties were selected to test antibiofilm properties.

The process of biofilm formation is affected by all honey samples against all bacteria. Antibiofilm activity was higher for Gram-negative bacteria (EC, ES and PA) compared to Gram-positive bacteria (SA and MR). For example, 53% to 65% of EC biofilm biomass development is inhibited by tested honey samples, but ES biofilm development is inhibited from 63% to 74%, and PA biofilm development is inhibited from 40% to 72%. The range of SA and MR biofilm biomass development inhibition by honey samples is broader, from 25% to 62% for SA and 24% to 63% for MR (Figure 3a).

##### The Activity of Honey Samples against Preformed Biofilm

All selected honey samples showed similar tendencies against biofilm development prevention—better activity against Gram-negative bacteria (EC and ES) biofilms than Gram-positive (SA and MR) bacteria. Activity against preformed biofilm of PA decreases when the range is from 3% to 72% for 24 h biofilms and from 17% to 78% for 48 h biofilms. All honey samples for established biofilms reduced more than 50% of the ES biofilm biomass after 24 h and 48 h. As *Staphylococcus* is known as a strong biofilm producer, the activity of honey samples against SA and MR biofilm biomass gets lower. Only a few honey samples can reduce biofilm biomass by more than 50% (Figure 3b,c).

The SEM analysis visualized the activity of the honey samples against the preformed biofilm, and results are similar to tests conducted in 96-well plates. All honey samples reduced more than 50% of EC and ES biofilm biomass. Less activity was seen against SA and MR when only a few samples (Hea_1, Hea_3 and Hor_1) reduced more than 50% of the biofilm biomass (Figure 4 and Appendix A).

### 2.3. Chemometric Characterisation

The PCA analysis for antimicrobial properties of honey samples vs. polyphenolic profile was performed using a two-component model with a total variance of 95% (PC1 68.3%, PC2 23.5%) (Figure 5).

Each honey sample received a score based on the well-diffusion method results to perform the PCA analysis. The inhibition radius interval for each bacteria received a score (of 0–3). The maximum score was 13, and the minimal score 0. As a result, we obtained five groups based on the antimicrobial activity: very high (nine samples), high (four samples), medium (five samples), low (thirteen samples) and none (eight samples).

The PCA plot shows that all Latvian honey samples are placed close to each other without forming separate clusters.

All honey samples with very high antimicrobial activity are distributed into two clusters. The three buckwheat samples, Buck_2, Buck_5 and Buck_4, are displayed in a separate cluster (Figure 5b). The other samples with very high activity overlap with samples without antibacterial activity (Figure 5c). Three honey samples with high activity (Clo_4, Clo_6 and Buck_1) are distributed near buckwheat samples with very high activity.

Polyphenol profile fingerprints of all honey samples were compared by analyzing chromatograms. For example, when comparing heather honey sample Hea_1 with very high antibacterial activity and Hea_2 and Hea_3 with medium antibacterial activity, no significant differences were observed (Figure 6a). The similar polyphenol profiles could explain the close location of these samples in the PCA area (Figure 5). Comparing the polyphenol profile fingerprints of buckwheat honey samples with very high antibacterial activity (Buck_2 and Buck_4) and clover honey Clo_4 with high antibacterial activity, the polyphenol profiles are very similar between the samples (Figure 6b). The similar polyphenol profiles could explain the proximity in the PCA diagram. However, comparing the polyphenol profile of two honey samples with very high antibacterial activity, Hea_1 and Buck_4, differences in the polyphenol profiles can be observed. The different fingerprints could explain why drugs with similar antibacterial activity can be found in entirely different clusters (Figure 6c).

## 3. Discussion

The main objectives of our study were to evaluate the antibacterial activity and phenolic profile of 40 monofloral honey samples in Latvia.

The novelty of our research is that it reports on the effect of monofloral Latvian honey with different botanical origins on biofilm development and against preformed biofilm.

The polyphenolic profile studies of honey samples were performed using the UHPLC-HRMS method. The polyphenolic profiles were included and quantified in μg/kg for one plant hormone, seven phenolic acids, eight flavonoids and one water-soluble vitamin. Fourteen polyphenols were detected in all samples.

Considering that the qualitative and quantitative content of polyphenols in honey depends on the geographical origin, we chose to compare the results for our country’s closest neighbors, Lithuania and Poland.

The studied polyphenols and their profiles are different, so we chose phenolic acids for comparison: chlorogenic acid, gallic acid, ferulic acid, p-coumaric acid, p-hydroxy benzoic acid and syringic acid. We observed similar concentrations of p-hydroxybenzoic and p-coumaric, and significantly lower concentrations of gallic, chlorogenic and syringic acid, as reported by Polish scientists Puścion-Jakubik, A. et al. [30] and Dżugan, M. et al. [31]. Similarly, low concentrations of gallic and chlorogenic acid were reported by Ramanauskiene, K. et al. [32]. Latvian honey had significantly higher ferulic acid concentrations than Polish honey samples.

Studies of antibacterial properties confirmed the previously stated assertion in the literature that honey has higher antibacterial activity against Gram-positive compared to Gram-negative bacteria [31,33]. In our study, we observed that inhibition zones for SA and MR bacteria are almost twice as large as for EC and ES bacteria in the well-diffusion method. Like Dżugan, M. et al., we found low MIC and MBC concentrations against SA bacteria. For example, the lowest was 6.25 (*v*/*v*) for Polish buckwheat honey and 5% (*w*/*v*) for most Latvian honey samples. Compared with the buckwheat honey tested in the current study, other Latvian honey samples, Lin_7, Clo_7, Rap_5 and Wil_6, showed high antibacterial potential.

In addition, studies in the literature indicate the activity of honey against *Candida albicans* infections [8]. However, our research still needs explicit confirmation regarding honey samples from Latvia.

Latvian honey’s antibacterial activity was compared with Manuka honey’s activity. Eleven honey samples exhibited higher inhibition zones on SA, ten on MR, ten on PA and two on EC and ES bacteria.

These findings indicate the potential use of Latvian honey in treating wound infections as well as their potential use when added to various biomaterials, as is widely the case with Manuka honey.

Until now, very little research has been performed on the ant-biofilm properties of Latvian honey. This work studied biofilm development prevention and honey samples’ activities against performed biofilms. Fifteen Latvian herbs which showed the best antibacterial activity were selected for the study.

All selected honey samples showed higher activity against Gram-negative bacteria biofilms (EC and ES) compared to Gram-positive bacteria biofilms (SA and MR).

The PCA analysis of the antimicrobial properties of the honey samples was performed. It showed that the antimicrobial activity of honey does not depend on its botanical origin. By assigning conditional scores to honey samples based on the inhibition radii obtained via the well-diffusion method, we obtained five groups based on antimicrobial activity: very high (nine samples), high (four samples), medium (five samples), low (thirteen samples) and none (eight samples). Honey that belonged to the “very high” activity group was divided into two clusters, one of which overlapped with the activity group—“none”. Therefore, we evaluated the polyphenolic content, concentrations and chromatographic fingerprints of the polyphenol profile. Although the literature studies have shown that individual phenolic (rutin, chlorogenic acid, etc.) compounds can have an inhibitory effect on biofilm formation, we observed that the activity and position of honey samples in the PCA cluster are most influenced not by the increased or decreased concentration of a single polyphenol compound, but by the synergy of several polyphenols, such as rutin, p-hydroxybenzoic acid, p-coumaric acid, kaempferol, chlorogenic acid and 3,4-dihydroxybenzoic acid [34,35]. The literature studies have shown that the antimicrobial action of phenolic acids, such as gallic, chlorogenic and p-coumaric acids, can decrease the extracellular pH, leading to an unfavorable environment for bacterial growth and can damage bacterial cell walls, leading to cell death [36,37,38,39].

To evaluate the chromatograms, we used the fingerprint method, comparing the total polyphenol profile, which helped us explain the honey sample’s position in the PCA cluster [40].

The studies showed the potential of using Latvian honey in wound healing. Com-pared to Manuka honey, it is much more difficult to obtain fully monofloral honey samples in Latvia because many plants bloom profusely during the honey collection season. This could cause problems obtaining honey with an identical composition several years in a row. The correlation of polyphenol analysis with antibacterial activity shows that it is necessary to study honey samples with a determined botanical origin for several seasons in a row and other honey parameters that could affect antibacterial activity, such as methylglyoxal composition, pesticides or very high or deficient concentrations of individual polyphenols.

## 4. Materials and Methods

### 4.1. Origin and Characterization of Honey Samples

Forty monofloral honey samples collected from beekeepers in the territory of Latvia were used, incuding 10 groups of different botanical orignin (Figure 7). Details of tested honey groups were shown in Table 4.

In all microbiology experiments, certified commercial Manuka honey (Produced by Oha Honey LP (Masterton, New Zealand), with a certified methylglyoxal content—800+ mg/kg) and two sugar analogue solutions were used for comparison. Sugar analogue solutions composed of 40% fructose, 30% glucose, 8% maltose, 2% sucrose and 45% fructose, 38% glucose and 1% of sucrose were used as the blank samples for determination of antimicrobial activity to mimic the concentration and composition of the main sugars in honey [41,42].

The honey samples were stored in the dark at room temperature before sample preparation.

### 4.2. Analytical Methods

The overall scheme of the research is shown in Figure 1.

#### 4.2.1. Melissopalynological Analysis

Pollen content in monofloral honey was determined via melissopalynology analysis to validate sample botanical origins. Samples were prepared according to the method of Louveaux et al. [43], with a total count and identification of 500 pollen grains per sample. The predominant pollen type was expressed as a percentage, and a sample was considered monofloral if it exceeded 45%. Within the legislation of Latvia [44], there are exceptions for predominant pollen percentage to rapeseed (*Brassica napus*) (>70%), heather (*Calluna vulgaris*) (>40%), buckwheat (*Fagopyrum esculentum*) (>25%) and linden (*Tilia cordata*) (>17%) floral origins. Thirty-nine samples exceeded legislation criteria, while sample Rap_5 had 69% rapeseed pollen. Due to the reasonably frequent presence of rapeseed pollen, and for nearly fulfilling the requirements, it was kept in the experiment sample set.

#### 4.2.2. Characterisation by pH

To determine pH, 1 g of honey was dissolved in 10 mL of ultrapure water (10% *w*/*v*), and the pH of the solution was measured (pH-meter SevenCompact S220, Mettler Toledo, Greifensee, Switzerland). Maximum and Minimum pH values can be found in Table 4.

#### 4.2.3. Characterization via UHPLC-HRMS Systems

The UHPLC-HRMS method was performed according to the previously developed analytical method by Rusko et al., to determine the polyphenol concentrations in monofloral honey samples [26]. A total of 0.5 g of honey sample was placed in a 2-mL Eppendorf tube and dissolved in 0.5 mL of 10% NaCl in a 0.01 M HCl (pH = 2) solution. The 1 mL of acetonitrile (MeCN) was added and vortexed for 1 min at 2000 rpm and centrifugated for 1 min at 15,000 rpm. The organic phase was collected in a 2-mL crimp-top chromatography vial. The 1 mL of MeCN was added to the water phase, and the extraction procedure was repeated until 1.9 mL of the organic phase was collected. The organic phase was dried under a gentle nitrogen flow at room temperature and reconstituted in a 0.5 mL deionized water/MeCN mixture (98:2 *v*/*v*), with 0.1% formic acid added to chromatography vials. Prepared samples were stored at 4 °C in the dark prior to the analysis.

The chromatographic separations were performed with the UHPLC system Dionex UltiMate 3000 (Thermo Scientific, Olten, Switzerland) equipped with a Krudkatcher™ in-line filter, 2.0-μm depth filter × 0.004 in i.d. and Kinetex PFP, 1.7 μm and 100 Å, 3.00 × 100 mm column (Phenomex, Torrance, CA, USA). The column thermostat was set to 40 °C, and the mobile phase flow rate was 0.6 mL/min. The autosampler sample storage was set to 4 °C, and the injection volume was 5 μL. The mobile phase consisted of (A) deionized water and (B) MeCN, both with 0.1% formic acid. The gradient conditions were 2.5 min pre-injection equilibration held at 2% B; 0–1 min, 2% B; 1–1.5 min, 2–25% B; 1.5–7 min, 25–60% B; 7–7.5 min, 60–98% B; 7.5–9.4 min, 98% B; 9.4–9.5 min, return to the initial 2% B. A diverter valve was used, and flow to the HRMS system was switched at 1.3 min.

The mass spectrometry analysis was performed using a Q Exactive system (Thermo Scientific, Bremen, Germany) equipped with a heated electrospray ionization (HESI) ion source. HESI conditions were as follows: negative ionization mode spray voltage: −3500 V; sheath gas: 40 a.u.; auxiliary gas: 10 a.u.; capillary temperature: 280 °C; heather temperature: 420 °C. The HRMS instrument was set to scan 100 at 1200 *m*/*z*. Thermo Scientific Xcalibur™ software v 4.1.31.9 was used for the quantification.

### 4.3. Antibacterial Activity Assay

#### 4.3.1. Bacterial and Fungal Strains

To evaluate the antibacterial properties of the honey samples, reference cultures of *Escherichia coli* (EC) ATCC 25922, *Extended-Spectrum Beta-Lactamases* (ES), *Pseudomonas aeruginosa* ATCC 27853 (PA), *Staphylococcus aureus* (SA) ATCC 25923 and *Methicillin-Resistant Staphylococcus aureus* (MR) were used in the study. The antifungal properties of the honey samples were identified against the reference culture of *Candida albicans* (CA) ATCC 10231. Both clinical isolates were previously isolated from pus and urine samples and identified with the VITEK2 system (bioMé-rieux, Marcy-l’Étoile, France). According to the European committee on antimicrobial susceptibility testing (EUCAST), the disc diffusion method was used to confirm bacterial resistance.

#### 4.3.2. The Antimicrobial Activity Using the Well-Diffusion Method

A modification of a standard disk diffusion test or Kirby–Bauer test, the well-diffusion method was used to test antimicrobial properties. Microbial suspensions were made with a densitometer (Biosan, Riga, Latvia) according to a 0.5 McFarland optical density. Sterile cotton swab suspensions were inoculated on a Mueller Hinton agar (MHA) (Oxoid, Oxoid-Hampshire, UK) plate, and four wells were made with ø 6 mm. Next, 60 µL of honey sample was added in each well to let it diffuse through the agar. MHA plates were incubated in a thermostat (Memmert, Schwabach, Germany) for 24 h at 37 °C. After incubation, the diameter of the inhibition zone around every well was measured.

#### 4.3.3. Determining of Minimum Inhibitory Concentration (MIC), Minimal Bactericidal Concentration (MBC) and the Minimum Fungicidal Concentration (MFC) with Broth Microdilution Method

The MIC and MBC values were investigated using the broth microdilution method, a standard laboratory antimicrobial susceptibility testing method. First, 10 mL of stock honey sample solution was prepared from an 80% honey solution (*w*/*v*) with Mueller Hinton broth (MHB) (Oxoid, UK). Next, 100 µL of the stock honey solution was seeded in a 96-well plate (SarsTEDT, Nümbrecht, Germany), and two-fold serial dilutions were performed in a 50 μL volume to reach a range between 40% and 0.312% for the honey samples. Finally, each well was seeded with 50 μL of microbial suspension (10^8^ CFU/mL) and were previously adjusted from suspensions of a 0.5 McFarland density. In the case of Candida albicans testing, a sabouraud dextrose broth (SDB) was used. All 96-well plates were wrapped with PARAFILM^®^ M (Merck, Darmstadt, Germany) and incubated in a thermostat (Memmert, Büchenbach, Germany) for 24 h at 37 °C. After incubation, absorbance values were measured with a microplate reader at 570 nm (Tecan Infinite F50, Männedorf, Switzerland). Absorbance levels were compared with the negative control, pure MHB without microorganisms, and the positive control, microorganisms without honey samples. MIC was considered as the lowest concentration where visual inhibition of microbial growth is observed in a 96-well plate.

To determine MBC and MFC, extra cultivation of 10 µL from wells on non-selective agar plates (Oxoid, UK) for bacteria and sabouraud dextrose agar (Oxoid, UK) for *Candida albicans* were performed in order—one above the MIC value and all remaining below the MIC value. Agar plates were incubated in a thermostat (Memmert, Büchenbach, Germany) for 18 h at 37 °C.

#### 4.3.4. Antibiofilm Activity—Prevention of Biofilm Development

A microtitre plate assay with crystal violet staining assay was used to evaluate the anti-biofilm activity of honey samples in the initial phase of biofilm development. In total, 200 μL of bacterial suspension (10^6^ CFU/mL) in a trypticase soy broth (TSB) (Oxoid, UK) was added into individual 96-well plates and incubated at 37 °C for 4 h. Following incubation, the plates were washed with 200 µL of sterile distilled water and air-dried. Afterwards, 200 µL of 80% (*w*/*v*) honey solution was added to individual wells and incubated for 24 h at 37 °C. A crystal violet staining assay was performed after incubation; washing of plates was repeated three times with sterile NaCl 0.9%. Next, the plates were stained with 200 µL 0,1% crystal violet stain for 15 min, after which the plates were washed three times with sterile distilled water. To remove the unabsorbed stain, 200 µL of ethanol was used, and absorbance was measured with a microplate reader at 570 nm (Tecan Infinite F50, Männedorf, Switzerland).

#### 4.3.5. Antibiofilm Activity—The Activity of Honey Samples against Preformed Biofilms

To evaluate the antibiofilm activity of honey samples against preformed biofilms, a microtiter plate assay with crystal violet staining assay was used, as described in Section 4.3.4. However, two series of 96-well plates were made—one was incubated at 37 °C for 24 h, but the second was cultivated at 37 °C for 48 h. Afterwards, 200 µL of 80% (*w*/*v*) honey solution was added to individual wells and incubated for 24 h at 37 °C. Following incubation, the plates were washed, stained with crystal violet, decolorized and measured with a microplate reader at 570 nm (Tecan Infinite F50, Männedorf, Switzerland) as described before.

#### 4.3.6. Evaluation of Antibiofilm Activity with Scanning Electron Microscopy (SEM)

Bacterial biofilms were cultured using a Calgary Biofilm Device (CBD) as described by Ceri et al., to determine the antibiofilm activity of honey samples via SEM [45]. Biofilms were prepared from a bacterial suspension of 1.0 after McFarland density and diluted 1 in 30 with sterile TSB. In total, 150 µL of diluted suspension was seeded in CBD and cultivated for 24 h at 37 °C with 150 rpm. After cultivation, biofilms on the polystyrene pegs of the CBD were rinsed three times in a sterile 0.9% NaCl solution and dipped in a freshly prepared 96-well plate with honey samples. The plates were incubated for 24 h at 37 °C. Next, the polystyrene pegs were rinsed three times in a sterile 0.9% NaCl solution. Afterwards, the polystyrene pegs were cut off, and biofilms were fixed with 2.5% glutaraldehyde for SEM. A scanning electron microscope Tescan Mira/LMU (Tescan, Brno, Czech Republic) was used to visualize the morphology of the obtained bacteria samples. Sputter-coated samples were examined at 5 kV using secondary electrons. Before analysis, samples were attached to aluminum pin stubs with conductive carbon tape, later sputter-coated with a thin layer of gold at 25 mA for 3 min using Emitech K550X (Quorum Technologies, Ash-ford, Kent, UK).

### 4.4. Chemometric and Data Analysis

PCA and HCA were performed using SIMCA 17 software (Umetrics, Umea, Sweden). The formation of clusters was visualized in scatter plots, dendrograms and loadings. HCA was calculated using Ward’s algorithm.

ANOVA and Dixon r10 outliner tests for polyphenols were performed using Minitab 17 Statistical software (Minitab, Brandon Court, UK).

## 5. Conclusions

Latvian honey has excellent potential as an agent in treating wounds due to its antibacterial properties. Monofloral honey samples with a high polyphenolic compound concentration show significant antimicrobial effects. The phenolic compound contents are partially responsible for honey’s antibacterial activity. The increased antimicrobial activity of the honey samples is based on several phenolic compound synergetic effects.

## Data Availability

Data are contained within the article.

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
