# Peer review of "Antimicrobial and Antibiofilm Properties of Latvian Honey against Causative Agents of Wound Infections"

_antibiotics, 2023, doi:10.3390/antibiotics12050816_

Round 1

Reviewer 1 Report

The topic of the manuscript (antibiotics-2351450-v1) is an interesting because honey is widely used in traditional medicine and modern wound healing biomaterial research. In the study, the authors evaluated the antibacterial activity and polyphenolics profiles of 40 monofloral honey samples collected from beekeepers in the territory of Latvia. The results showed Latvian honey possesses a promising potential to be used in wound healing biomaterials. Regarding results, some concepts should be carefully revised. There are many problems that have not been explained clearly.

1) The quality of figures should be improved.

2) Chlorogenic acid is a sensitive substance. How dose chlorogenic acid work in biofilms? (line 290)

3) Is honey the main antibacterial substance in the biofilms? If polyphenolics are the main active substance, how about the effect of using a single compound?

4) What to do with sugar from honey in the biofilms?

Minor editing of English language required

Author Response

Dear reviewer,

We would like to thank you for your constructive comments on our manuscript (antibiotics-2351450), which is incredibly beneficial to improve the quality of this research article. As shown below, we have addressed the concerns and answered all the questions in our point-to-point response.

We have carefully revised the manuscript and marked the changed text in red.

Point 1 The quality of figures should be improved.

Response 1: We thank the reviewer for this valuable suggestion.

We improved the quality of Figures.

For example, in Figure 2, we left only two examples (for SA nad MR bacteria) with inhibition zone diameters; the other diagrams we included as supplementary material Figure S1.

For Figure 3, we made a statistical analysis.

Point 2: Chlorogenic acid is a sensitive substance. How dose chlorogenic acid work in biofilms? (line 290)

Response 2: Thank you for your comments and valuable suggestions. We have added content on chlorogenic acid; in addition, we added information on the antimicrobial action of other phenolic acids, like gallic and p-coumaric acid.

Please see lines 339 - 348.

Although literature studies have shown that individual phenolic (rutin, chlorogenic acid, etc.) compounds can have an inhibitory effect on biofilm formation, we observed that the activity and position of honey samples in the PCA cluster are most influenced not by the increased or decreased concentration of a single polyphenol compound but by the synergy of several polyphenols, like rutin, p-hydroxybenzoic acid, p-coumaric acid, kaempferol, chlorogenic acid and 3,4 - dihydroxybenzoic acid [30,31]. Literature studies have shown that the antimicrobial action of phenolic acids, such as gallic, chlorogenic and p-coumaric acids, can decrease the extracellular pH, leading to an unfavorable environment for bacterial growth and can damage bacterial cell walls, leading to cell death [32-35].

Point 3: Is honey the main antibacterial substance in the biofilms? If polyphenolics are the main active substance, how about the effect of using a single compound?

Response 3: We thank the reviewer to draw our attention to the complex composition of honey.

Please find information about complex honey composition lines 49-55.

Although the main constituents of honey are reducing carbohydrates, the composition of honey is very complex, and it contains about 200 substances [9].

Lines 58-69 we added information on, polyphenol action mechanisms of antimicrobial action.

Polyphenols are a group of natural compounds that are widely distributed in plants and have been found to possess antibacterial properties against a range of Gram-positive and Gram-negative bacteria. Various mechanisms of antimicrobial action of polyphenols have been described in the literature, e.g. disrupting the bacterial cell membrane (pore formation, disintegration of membrane proteins, cell wall disruption, modifying membrane potential), affecting cytoplasm (leaking cell components, cytoplasm acidification, chelation of metal ions), and disturbing functions (DNA/RNA/protein synthesis, modulate a cellular redox response through a proline-linked pentose phosphate pathway, inactivation of enzymes, loss of biofilm formation) [12,13]. Similar to honey, polyphenols also have also been shown to enhance the activity of antibiotics against biofilms of several bacterial species, including Pseudomonas aeruginosa (PA) and Staphylococcus aureus (SA) [14].

Discussion part lines: 339-345

Although literature studies have shown that individual phenolic (rutin, chlorogenic acid, etc.) compounds can have an inhibitory effect on biofilm formation, we observed that the activity and position of honey samples in the PCA cluster are most influenced not by the increased or decreased concentration of a single polyphenol compound but by the synergy of several polyphenols, like rutin, p-hydroxybenzoic acid, p-coumaric acid, kaempferol, chlorogenic acid and 3,4 - dihydroxybenzoic acid [30,31].

Information on individual polyphenols e.g. phenolic acids can be found lines 345-347

Literature studies have shown that the antimicrobial action of phenolic acids, such as gallic, chlorogenic and p-coumaric acids, can decrease the extracellular pH, leading to an unfavorable environment for bacterial growth and can damage bacterial cell walls, leading to cell death [32-35].

Point 4: What to do with sugar from honey in the biofilms?

Response 4: Thank you for your comments.

We add additional information regarding sugar content and interaction with biofilms in the introduction part, lines 51-55:

The high sugar content and viscosity of honey have been shown to play a crucial role in inhibiting microbial growth and preventing the formation of biofilms. Furthermore, the biofilm matrix is composed of polysaccharides, and recent studies have shown that sugar molecules are also utilized as chemical messengers between bacterial species within the biofilm structure [10].

Please see information about in the section Materials and Methods about blank experiments, lines 403-408:

“In all microbiology experiments, certified commercial Manuka honey (Produced by Oha Honey LP, with certified methylglyoxal content – 800+ mg/kg) and two sugar analogue solutions were used for comparison. Sugar analogue solutions composed of 40% of fructose, 30% of glucose, 8% of maltose, 2% of sucrose and 45% fructose, 38% glucose and 1% of sucrose were used as the blank samples for determination of antimicrobial activity to mimic the concentration and composition of the main sugars in honey”.

Reviewer 2 Report

The authors describe their work to investigate the antimicrobial properties of Latvian honey. The honey was analyzed using polyphenolic profiles as well as antibacterial effect with both gram positive and negative bacteria. Overall the paper was well written and complete. The authors supplied important data and drew appropriate conclusions. I see no reason for additional changes to be made. 

Author Response

Dear reviewer,

We would like to thank you for your constructive comments on our manuscript (antibiotics-2351450).

Reviewer 3 Report

The current manuscript is interesting to read, focusing on the characterization of Latvian honey, and assessment of its antimicrobial properties. Nevertheless, some issues should be addressed before acceptance for publication:

- The wound healing properties of the honey were not assessed, correct? Only the antimicrobial properties. Although these could be useful for infected wounds, it does not show specific wound healing properties, which would imply tissue regeneration. Hence, it is misleading to write in this way, please correct;

- In figure 2 graphs should be bigger, since it is hard to read the axis; same with figure 3; also statistical analysis should be added, if possible;

- Please compare the obtained results, in what concerns antimicrobial properties and chemical composition, to honey from other origins (other countries);

- Please add a comment on how the use of honey could help combat antibiotic resistance.

Author Response

Dear reviewer,

We would like to thank you for your constructive comments on our manuscript (antibiotics-2351450), which is incredibly beneficial to improve the quality of this research article. As shown below, we have addressed the concerns and answered all the questions in our point-to-point response.

We have carefully revised the manuscript and marked the changed text in red.

Point 1 The wound healing properties of the honey were not assessed, correct? Only the antimicrobial properties. Although these could be useful for infected wounds, it does not show specific wound healing properties, which would imply tissue regeneration. Hence, it is misleading to write in this way, please correct;

Response 1: We thank the reviewer for this important point.

We have corrected the paper's title to "Antimicrobial and antibiofilm properties of Latvian honey against causative agents of wound infections", thus emphasizing exactly the antimicrobial properties.

Point 2 In figure 2 graphs should be bigger, since it is hard to read the axis; same with figure 3; also statistical analysis should be added, if possible;

Response 2: We thank the reviewer for this valuable suggestion.

We improved the quality of Figures 2 and 3. In Figure 2, we left only two examples with inhibition zone diameters; the other diagrams we included as supplementary material.

For Figure 3, we made a statistical analysis.

Point 3 Please compare the obtained results, in what concerns antimicrobial properties and chemical composition, to honey from other origins (other countries);

Response 3: Thank you for your comments and valuable suggestions, lines 295-305.

Considering that the qualitative and quantitative content of polyphenols in honey depends on the geographical origin, we chose to compare the results for our country's closest neighbors, Lithuania and Poland.

The studied polyphenols and their profiles are different, so we chose phenolic acids for comparison: chlorogenic acid, gallic acid, ferulic acid, p-coumaric acid, p-hydroxy benzoic acid and syringic acid. We observed similar concentrations of p-hydroxybenzoic, p-coumaric and significantly lower concentrations of gallic, chlorogenic and syringic acid as reported by Polish scientists Puścion-Jakubik, A. et al. [30] and Dżugan, M. et al. [31]. Similarly, low concentrations of gallic and chlorogenic acid were reported by Ramanauskiene, K. et al. [32]. Latvian honey had significantly higher ferulic acid concentrations than Polish honey samples.

And lines: 310 – 314.

Like Dżugan, M. et al., we found low MIC and MBC concentrations against SA bacteria. For example, the lowest was 6.25 (v/v) for Polish buckwheat honey and 5% (w/v) for most Latvian honey samples. Compared with the buckwheat honey tested in the current study, other Latvian honey samples Lin_7, Clo_7, Rap_5, Wil_6 showed high antibacterial potential.

Point 4 Please add a comment on how the use of honey could help combat antibiotic resistance.

Response 4: Thank you for your comment.

We added a comment on honey’s synergetic effect with antibiotics on biofilms in the lines 44-49

Several studies recommend using honey directly or together with antibiotics because it increases the antimicrobial effect, and creates a synergistic effect or additive interaction against bacterial biofilms [6,7]. Using honey alone or together with antibiotics could reduce the use of antibiotics, minimise adverse antibiotic reactions and increase the effectiveness of treatment [8].

Round 2

Reviewer 1 Report

The authors have great  job.

Author Response

We thank the reviewer for your comments.